# Tailoring COVID-19 Vaccination Strategies in High-Seroprevalence Settings: Insights from Ethiopia

**DOI:** 10.3390/vaccines12070745

**Published:** 2024-07-05

**Authors:** Esayas Kebede Gudina, Kira Elsbernd, Daniel Yilma, Rebecca Kisch, Karina Wallrafen-Sam, Gemeda Abebe, Zeleke Mekonnen, Melkamu Berhane, Mulusew Gerbaba, Sultan Suleman, Yoseph Mamo, Raquel Rubio-Acero, Solomon Ali, Ahmed Zeynudin, Simon Merkt, Jan Hasenauer, Temesgen Kabeta Chala, Andreas Wieser, Arne Kroidl

**Affiliations:** 1Department of Internal Medicine, Institute of Health, Jimma University, Jimma P.O. Box 378, Ethiopia; daniel.yilma@ju.edu.et; 2Institute of Infectious Diseases and Tropical Medicine, LMU University Hospital, LMU Munich, 80802 Munich, Germany; kira.elsbernd@lrz.uni-muenchen.de (K.E.); rebecca.kisch@med.uni-muenchen.de (R.K.); raquel.rubio@med.uni-muenchen.de (R.R.-A.); wieser@mvp.lmu.de (A.W.); akroidl@lrz.uni-muenchen.de (A.K.); 3Institute for Medical Information Processing, Biometry, and Epidemiology, LMU Munich, 81377 Munich, Germany; 4Life and Medical Sciences (LIMES), University of Bonn, 53115 Bonn, Germany; karina.wallrafen@uni-bonn.de (K.W.-S.); simon.merkt@uni-bonn.de (S.M.); jan.hasenauer@uni-bonn.de (J.H.); 5School of Medical Laboratory Sciences, Institute of Health, Jimma University, Jimma P.O. Box 378, Ethiopia; gemeda.abebe@ju.edu.et (G.A.); zeleke.mekonnen@gmail.com (Z.M.); ayyurayyan@gmail.com (A.Z.); 6Department of Pediatrics and Child Health, Jimma University, Jimma P.O. Box 378, Ethiopia; melkamuarefayine@gmail.com; 7Data Synergy and Evaluation, African Population and Health Research Center, Nairobi 00100, Kenya; mjebena@aphrc.org; 8Jimma University Laboratory of Drug Quality (JuLaDQ), and School of Pharmacy, Institute of Health, Jimma University, Jimma P.O. Box 378, Ethiopia; sultan.sulemanl@gmail.com; 9Tropical Health and Education Trust, Addis Ababa P.O. Box 1165, Ethiopia; yoseph.mamo@thet.org; 10Saint Paul’s Hospital Millennium Medical College, Addis Ababa P.O Box 1271, Ethiopia; solali2005@gmail.com; 11Institute of Computational Biology, Helmholtz Zentrum München—German Research Center for Environmental Health, 85764 Neuherberg, Germany; 12Department of Health Policy and Management, Institute of Health, Jimma University, Jimma P.O. Box 378, Ethiopia; ktemesgen@yahoo.com; 13German Center for Infection Research (DZIF), Partner Site Munich, 80802 Munich, Germany; 14Immunology, Infection and Pandemic Research IIP, Fraunhofer ITMP, 80802 Munich, Germany; 15Faculty of Medicine, Max von Pettenkofer Institute, LMU Munich, 81377 Munich, Germany

**Keywords:** cost-effectiveness, COVID-19 vaccine, Ethiopia, low-income setting, hybrid immunity, SARS-CoV-2 infection

## Abstract

This study aimed to retrospectively assess the cost-effectiveness of various COVID-19 vaccination strategies in Ethiopia. It involved healthcare workers (HCWs) and community participants; and was conducted through interviews and serological tests. Local SARS-CoV-2 variants and seroprevalence rates, as well as national COVID-19 reports and vaccination status were also analyzed. A cost-effectiveness analysis was performed to determine the most economical vaccination strategies in settings with limited vaccine access and high SARS-CoV-2 seroprevalence. Before the arrival of the vaccines, 65% of HCWs had antibodies against SARS-CoV-2, indicating prior exposure to the virus. Individuals with prior infection exhibited a greater antibody response to COVID-19 vaccines and experienced fewer new infections compared to those without prior infection, regardless of vaccination status (5% vs. 24%, *p* < 0.001 for vaccinated; 3% vs. 48%, *p* < 0.001 for unvaccinated). The cost-effectiveness analysis indicated that a single-dose vaccination strategy is optimal in settings with high underlying seroprevalence and limited vaccine availability. This study underscores the need for pragmatic vaccination strategies tailored to local contexts, particularly in high-seroprevalence regions, to maximize vaccine impact and minimize the spread of COVID-19. Implementing a targeted approach based on local seroprevalence information could have helped Ethiopia achieve higher vaccination rates and prevent subsequent outbreaks.

## 1. Introduction

The rapid development of SARS-CoV-2 vaccines has played a crucial role in mitigating the COVID-19 pandemic by significantly reducing severe illness, hospitalizations, and deaths [1,2]. During the first year of the pandemic, both previous infection [3,4] and vaccinations [5,6] were thought to provide effective protection against SARS-CoV-2 infection or reinfection. However, the emergence of immune escape variants, such as delta (B.1.617.2) and omicron (B.1.1.529), has led to an increase in vaccine breakthrough infections and reinfections, although available vaccines still offer substantial protection against severe disease [7,8]. Consequently, many countries have recommended additional vaccine doses to enhance protection against SARS-CoV-2 [9,10].

More than three years since the approval of the vaccines, only around 39% of people living in Africa have received at least one dose [11], primarily due to limited access to the vaccines [2] and vaccine hesitancy [12]. Interestingly, seroepidemiologic studies have shown that the majority of the population in Africa had already been exposed to the virus even before vaccines became available [13]. While countries have implemented various strategies to optimize vaccine use by prioritizing high-risk and vulnerable population groups [14], there is no clear vaccination strategy in settings with very high SARS-CoV-2 seroprevalence. There is, however, growing evidence suggesting that individuals previously infected with SARS-CoV-2 mount stronger immune responses after vaccination [15,16], resulting in improved protection attributable to “hybrid immunity”. Therefore, in settings with high SARS-CoV-2 seroprevalence, it is reasonable to reserve vaccine doses for the most vulnerable groups and provide less frequent doses to the general population. Given the shortage of SARS-CoV-2 vaccines and high SARS-CoV-2 seroprevalence in many African countries, context-specific vaccine utilization, rather than adopting strategies from other settings, is imperative for reducing severe disease burden quickly and in a cost-effective manner.

Ethiopia received 2.2 million doses of the Covishield COVID-19 vaccine on 7 March 2021 through the COVAX Facility [17]. Covishield is a recombinant ChAdOx1-S vaccine similar to the Oxford-AstraZeneca vaccine, manufactured by the Serum Institute of India [18]. On 13 March 2021, coinciding with the first anniversary of the country’s initial COVID-19 report, Ethiopia began vaccinating its healthcare workers (HCWs) as a priority population group. The original plan was to receive an additional 5.4 million doses by May 2021 and vaccinate all HCWs and at least 20% of the general population by the end of 2021 [19]. However, the import of more doses was halted due to India’s decision to prioritize its own population after experiencing a major COVID-19 outbreak in April 2021 [20]. This resulted in a critical vaccine shortage in Ethiopia, causing a delay in the COVID-19 vaccination campaign for the general population until November 2021 [21]. Consequently, by the end of 2021, only around 9.3 million people (<8% of the population) had received at least one dose of the vaccine [11].

Ethiopia implemented a vaccination strategy that involved either two doses of the Covishield vaccine, with a three-month gap between doses, or a single dose of the Johnson & Johnson vaccine. Due to the high seroprevalence of SARS-CoV-2 [22] and limited vaccine availability in the country, a pragmatic vaccination strategy based on the local context could have maximized the impact of the vaccine by prioritizing it for the most vulnerable groups. In this study, we aimed to determine the cost-effectiveness of vaccination strategies in Ethiopia in the context of the timing of available vaccine doses, pre-vaccination seroprevalence, and post-vaccination antibody response. We based our analysis on data obtained from the antibody responses after one dose of the AstraZeneca/Covishield vaccine, local SARS-CoV-2 variant data, local SARS-CoV-2 seroprevalence, national infection rate, and vaccination status.

## 2. Materials and Methods

### 2.1. Study Settings and Period

This study was conducted at the Jimma Medical Center (JMC) located in Jimma Town, southwest Ethiopia. The center serves as a university hospital with a staff of over 3000 HCWs. The first round of vaccination campaigns at the hospital started on 7 April 2021. The vaccination strategy was adopted from the World Health Organization interim recommendations for use of the AZD1222 (ChAdOx1-S) AstraZeneca COVID-19 vaccine [23]. As per this guideline, two doses of Covishield vaccine (COVISHIELD™, Serum Institute of India, Pune, India), with the second dose of the vaccine planned after twelve weeks, were used as a full vaccination strategy. However, access to the second dose of the vaccine was delayed due to the global shortage of the vaccine [21].

### 2.2. Study Participants

The study randomly selected and enrolled HCWs employed at JMC who were registered to receive the first dose of the COVID-19 vaccine during the first round of the vaccination campaign at JMC (7 to 12 April 2021). A follow-up post-vaccination visit was planned from 23 June to 13 July 2021, corresponding to the second phase of vaccinations. However, due to the delayed arrival of the second vaccine dose [20], participants were contacted by phone to join the subsequent phase, held between 20 June and 28 August 2021.

Community participants from our previous SARS-CoV-2 seroepidemiologic cohorts in Jimma, who were recruited during the same time period and had not received the COVID-19 vaccine, were selected as controls. These participants were part of a larger community cohort for a SARS-CoV-2 serosurvey in Ethiopia, and their data had previously been analyzed as part of the larger dataset [22,24].

### 2.3. Data Collection and Procedures

We utilized primary data from HCWs and community participants, as well as secondary data from SARS-CoV-2 RT-PCR-positive samples on circulating SARS-CoV-2 strains and national COVID-19 figures from the Ministry of Health of Ethiopia.

Primary data, including demographic information, previous COVID-19-related symptoms, history of COVID-19 diagnosis and severity of the disease, and history of clinical conditions were collected at the baseline pre-vaccination visit through interviews using structured questionnaires. During the post-vaccination visit, data about adverse events of the first dose of the COVID-19 vaccine and their severity, new-onset of COVID-19-related symptoms, recent COVID-19 diagnosis and severity, contact with COVID-19 confirmed or suspected persons, and any new-onset medical conditions were collected.

For SARS-CoV-2-specific serology, 2–3 mL of venous blood was collected using serum separator tubes from each study participant at both pre- and post-vaccination visits. Serum specimens were processed daily and stored at –20 °C. A semiquantitative anti-nucleocapsid antibody [Ro-N-Ig] test was performed at JMC Clinical Chemistry Laboratory with Elecsys^®^ Anti-SARS-CoV-2 assay using the Cobas 6000 module e601 system (Roche Diagnostics, Basel, Switzerland). Anti-SARS-CoV-2 anti-spike [Ro-RBD-Ig-quant] was measured using the Cobas e801 analytical unit (Roche Diagnostics, Basel, Switzerland) at the Ludwig Maximilian University of Munich (LMU), Germany. Both tests were performed according to the manufacturer’s instructions.

The results of the anti-nucleocapsid (anti-N) assay were interpreted based on semiquantitative measurements presented as cut-off index (COI). A COI value of ≥1 was interpreted as reactive (positive) for SARS-CoV-2 anti-N antibodies. The anti-spike (anti-S) antibody result was presented as binding antibody units (BAU/mL) with a linear range of 0.4 to 250 U/mL. A BAU value of ≥0.8 U/mL was considered as evidence for receptor-binding domain-specific (RBD) antibodies. Values above 250 U/mL were diluted until the linear range was reached according to the manufacturer’s instructions. Values in this study were measured within the linear ranges and back-calculated depending on the dilution factor, as appropriate.

Local SARS-CoV-2 variants were analyzed from 419 RT-PCR-positive nasal swabs collected from patients at JMC between January and September 2021. Samples were shipped to LMU for viral genomic sequencing as part of a larger COVID-19 survey study conducted at JMC and St. Paul’s Hospital in Addis Ababa [24].

National SARS-CoV-2 infection and COVID-19 vaccination rates in Ethiopia were retrieved from the daily official Facebook report of the Ministry of Health of Ethiopia (https://www.facebook.com/EthiopiaFMoH, accessed on 26 December 2023).

### 2.4. Data Management and Analysis

The primary data were double-entered into EpiData Manager, version 4.6.0.0, and then exported as a comma-separated values (CSV) file for analysis in R, version 4.3.1. Figures were created with BioRender.com. Participant baseline characteristics and reported adverse events were described as frequency and proportion for discrete data and mean and standard deviation or median and interquartile range (IQR) for continuous data. The antibody data were transformed using the shifted logarithm base 10 function (log10(x + 1)), enabling the comparison of results across different scales while still maintaining the interpretation of zero as the absence of detectable antibodies. Emerging COVID-19 infections were calculated as a change in anti-N over time (pre- and post-vaccination measurement) greater than two standard deviations of the Roche anti-N measurement deviation calculated using repeated measurements from the same sample from a previous serosurvey cohort [25]. An anti-N increase was thus interpreted as a SARS-CoV-2 infection in the absence of possible confounding whole inactivated virus vaccines administered.

The national test positivity rate, expressed as a percentage, the frequency of critical cases, and the proportions of locally circulating SARS-CoV-2 variants were described over the same time period as that of the primary data collection.

For the cost-effectiveness analysis, the cost per percentage point increase in the adult population of Ethiopia with anti-spike levels above 275 BAU/mL (relative to a no-vaccination scenario) [24] was assessed for three vaccination strategies: (a) giving two vaccine doses per person until vaccines run out; (b) giving one dose per person until vaccines run out; and (c) performing an antibody test on each vaccine-eligible individual and administering one dose if their test indicated a previous infection and two doses if not. In addition to our data on antibody responses to the AstraZeneca/Covishield vaccine, we used estimates of antibody responses to the Pfizer vaccine from Wheeler et al. [26] and estimates of vaccine costs from McDonnell et al. [27]. Antibody response to one exposure to the virus and one dose of a given vaccine was assumed to be similar to the response to two doses of that vaccine. The mathematical derivations for the cost-effectiveness analysis, the complete set of parameters, and the results for further health benefit metrics are provided in the Appendix A (Cost Effectiveness Analysis of SARS-CoV-2 Vaccination Strategies in Ethiopia).

## 3. Results

### 3.1. Baseline Characteristics of Study Participants

For the assessment of post-vaccine antibody response, a total of 254 HCWs (vaccinated group) and 81 community participants (unvaccinated group) were included in the final analysis. The baseline assessment for the vaccinated group took place during the week of 7 to 12 April 2021. A total of 259 of these participants attended the post-vaccination visit between 20 June and 28 August 2021, of whom 5 were finally excluded because they did not receive the vaccine. For the unvaccinated group (control), the respective observational periods extended from 1 February to 16 March and from 3 August to 7 September 2021 (Figure 1).

Baseline characteristics for vaccinated and unvaccinated cohorts are shown in Table 1 and Appendix A. Nearly two-thirds of the vaccinated group and about 40% of the unvaccinated group had evidence of previous SARS-CoV-2 infection (i.e., reactive anti-N) at baseline. Vaccine adverse effects were reported to be mild and non-life threatening (Appendix A).

The pre-vaccination assessment was conducted during the second wave of the COVID-19 outbreak in Ethiopia, which was characterized by a high RT-PCR test positivity rate (15.4% to 28.6%) and the highest reported absolute number of critical cases resulting in intensive care unit (ICU) admissions. The dominant virus variant during this time was the alpha variant (B.1.1.7). The post-vaccination follow-up was performed after the end of the outbreak caused by the alpha variant and at the beginning of the delta variant (B.1.617.2), which likely triggered the third wave of COVID-19 outbreak in Ethiopia (Figure 2).

### 3.2. Post-Vaccine Antibody Response and Rate of SARS-CoV-2 Reinfection

Among vaccinated participants, Anti-S antibody titers were higher in the post-vaccination period compared to the pre-vaccination period (Figure 3A,B). Anti-S antibody titers in the post-vaccination period were highest among those with prior infection who were vaccinated. The median post-vaccination Anti-S antibody titer of those vaccinated without prior infection was comparable to the pre-vaccination level of those participants with prior infection (42.5 vs. 40.6 U/mL), indicating a comparable increase in Anti-S antibody titer following vaccination and infection. Individuals who were vaccinated and had a previous SARS-CoV-2 infection experienced a significantly greater increase in both Anti-S and Anti-N as a surrogate of protective immunity as compared to those who had not been previously infected and received vaccination (*p* < 0.001) (Figure 3C).

In terms of the frequency of SARS-CoV-2 infection during the follow-up period, individuals who had prior infections had fewer new infections compared to those who had no prior infection, regardless of their vaccination status (5% vs. 24%, *p* < 0.001 for vaccinated participants; 3% vs. 48%, *p* < 0.001 for unvaccinated individuals). Among those who had no prior infection, participants who did not receive vaccination had a higher infection rate compared to those who received vaccination (48% vs. 24%, *p* = 0.007). This lesser immune protection is also reflected among unvaccinated participants, as those with no prior infection had a higher rate of infection during the study period compared to those who had a prior infection (48% vs. 3%, *p* < 0.001, Figure 3D).

### 3.3. Cost-Effectiveness of Vaccine Strategies

Based on these findings, we conducted an analysis to determine the most cost-effective vaccination strategy in settings similar to Ethiopia, where vaccines were scarce and SARS-CoV-2 seroprevalence was high. This approach would allow for dose sparing by administering a single vaccine dose as a booster for seropositive individuals, while providing a two-dose vaccination strategy for seronegative individuals. We used SARS-CoV-2 Anti-N antibody positivity as a surrogate for pre-existing natural infection before vaccination; as such, we assumed an initial infection prevalence of 61.3% based on our data.

For the AstraZeneca (Covishield) vaccine, with a per-dose cost of USD 3 [27], the cost per percentage point increase in the proportion of the adult population in Ethiopia with immunity was minimized for the single-dose strategy unless individual antibody testing (with perfect sensitivity and specificity) could be performed for less than USD 0.63 per test (Figure 4A). For the more effective but more expensive Pfizer vaccine, with a per-dose cost of USD 17 [27], our analysis showed that a single-dose strategy was optimal unless individual antibody testing, again with perfect sensitivity and specificity, could be performed for less than USD 2.10 per test (Figure 4B). These cost thresholds decreased when the assumption of perfect test sensitivity was relaxed. For example, if the test sensitivity was 95% instead of 100%, then the single-dose strategy would be optimal unless the per-test cost was less than USD 0.54 for AstraZeneca or less than USD 1.57 for Pfizer.

For both vaccines, these results are all sensitive to the prevalence of previous infection in the population for both vaccines: the two-dose strategy may be preferable to the single-dose strategy if the level of exposure to the virus is low enough at the start of the vaccine rollout (Figure 4C,D). For more details, including versions of Figure 4 for varying test sensitivities and a corresponding analysis optimizing the cost per log-unit increase in the population mean anti-spike level, refer to Appendix A under Cost Effective Analysis of SARS-CoV-2 Vaccination Strategies in Ethiopia.

## 4. Discussion

Nearly two-thirds of HCWs at Jimma Medical Center, a major referral center in Ethiopia, had anti-SARS-CoV-2 antibodies prior to the arrival of COVID-19 vaccines in April 2021. This suggests that a significant portion of the population had already been exposed to the virus by the time Ethiopia began its vaccination campaign (Figure 5). In our cohort study, individuals who had been infected with SARS-CoV-2 and received one dose of the Covishield vaccine had better anti-S antibody responses compared to those who had no evidence of SARS-CoV-2 infection at baseline and received one dose of the Covishield vaccine. This translated into a significantly higher SARS-CoV-2 infection rate for those who only received an incomplete vaccination regimen without prior natural immune priming.

When comparing the previously infected participant groups with respect to their vaccination status, no clinical benefit in terms of subsequent reinfection was observed between the vaccinated and unvaccinated groups; however, anti-S antibody responses were higher in the vaccinated participants, potentially indicating a higher impact on protective antibody durability and protection against severe courses. Considering that the peak of severe COVID-19 cases in Ethiopia occurred during the delta wave in 2021, as in many other African countries, a broader rollout of COVID-19 vaccines during the first half of 2021 could have mitigated the clinical impact at that time [24].

COVID-19 vaccines arrived in Africa later than the rest of the world. Ethiopia received the first batches of the vaccine only in March 2021. By that time, even though about 200,000 COVID-19 cases (less than 1% of the Ethiopian population) had been officially reported, nearly two-thirds of the HCWs included in this study had already been exposed to the virus. Studies conducted among HCWs and the general community in various parts of the country also revealed a high prevalence of SARS-CoV-2 in Ethiopia by early 2021 [22,28]. Despite the high infection rates and critical shortage of the vaccine, Ethiopia attempted to adopt a vaccination strategy of two doses (or one dose of Johnson & Johnson) for all individuals aged 12 years and older, following the approach used in most high-income countries [29,30].

However, a new wave of COVID-19 outbreaks, particularly in India, and the emergence of the delta variant led to a decline in global vaccine supply [20]. As a result, Ethiopia only managed to vaccinate about 8% of its population by the end of 2021, falling short of the 20% target (Figure 5). Furthermore, three years since the vaccines became available, only about 43% of the Ethiopian population have received at least one dose of the vaccine [11]. During these times, the country faced two major COVID-19 outbreaks: one from August to October 2021 attributed to the delta variant, and another from December 2021 to January 2022 attributed to the omicron variant [24].

Nevertheless, the number of deaths and severe/critical cases did not surge parallel with the number of COVID-19 cases since the end of April 2021, probably due to the underlying immunity in the population reflected by the high seroprevalence [22,24,28]. As described in previous studies [15,16], our study also revealed that individuals with previous SARS-CoV-2 infection who received one dose of the vaccine had higher antibody titers and lower rates of new SARS-CoV-2 infection compared to those with no prior infection. This demonstrates the role of hybrid immunity in reducing the risk of vaccine breakthrough infection and preventing severe diseases.

This evidence could have helped revise the vaccination strategy to align with the local context. Countries facing such a critical shortage of COVID-19 vaccines, such as Ethiopia or the entire WHO-Africa region, could have benefited from adopting more pragmatic vaccination strategies designed for areas with high SARS-CoV-2 seroprevalence. If a tailored vaccination strategy had been implemented earlier, it could have potentially saved more vaccine doses, maximized vaccine impacts on a population level, and even reduced vaccine hesitancy.

In our cost-effectiveness analysis, we therefore evaluated a strategic alternative of SARS-CoV-2 Anti-N screening for targeted vaccine utilization. Our cost-effectiveness analysis indicates that in the case of limited vaccine availability, the relative health value for money of different vaccination strategies depends on the prevalence of previous infection. Hence, surveillance testing is necessary to determine the optimal strategy for a specific region. If the level of exposure to the virus is moderate or high, our analysis suggests that it would be advisable to administer a single dose of the vaccine per person. This is particularly important in high-burden, low-resource settings, where it is crucial to reach as many people possible with the vaccine, even if they cannot receive the recommended two doses as a result. Point-of-care Anti-N screening as part of an individualized vaccine approach was a cost-effective option only if per-test costs were lower than USD 0.63 with Covishield or lower than USD 2.10 with mRNA vaccines such as Pfizer.

Ethiopia scaled up its COVID-19 vaccination efforts in February 2022, nearly a year after starting vaccination program for HCWs. Consequently, nearly 80% of the total vaccine doses have been administered, and about 50% of the population who have received at least one dose of the vaccine in Ethiopia were vaccinated only since February 2022. At this time, the major waves of the pandemic in Ethiopia were almost over, as indicated by a decrease in the number of new COVID-19 cases and a population-level SARS-CoV-2 seroprevalence rate of 95% (Figure 5). This suggests that the vaccination strategy used by the country may not have had as much impact on the outbreak pattern as intended [24].

Our approach integrated newly generated data from post-vaccine antibody response analysis, a SARS-CoV-2 serosurvey, and the sequencing of SARS-CoV-2 samples collected during the study period. We compared these findings with the national COVID-19 outbreak and vaccination trends, enhancing the robustness of our methodology. Conducting the post-vaccine antibody response analysis between two significant COVID-19 waves in Ethiopia, caused by the alpha and delta variants, allowed us to realistically reflect the on-ground situation.

Although the data used in this study are already a few years old, and thus are unlikely to affect approaches for COVID-19 vaccination now, they provide an important insight into the fact that adopting strategies proven to be effective in other settings may be counterproductive. Such shortcomings can only be addressed by enhancing local evidence generation and synthesis to guide policy, something that has been in short supply in Africa during the pandemic, and still is not on par with many other regions of the globe [31].

Other limitations of our study include the single post-vaccine antibody assessment conducted three months after the first dose and before the emergence of most immune escape variants like omicron. Previous SARS-CoV-2 infections were identified using the Elecsys^®^ anti-SARS-CoV-2 anti-nucleocapsid antibody test (Roche diagnostics), which may fail to detect a certain fraction of infections that occurred over a year prior due to signal loss. This test also does not distinguish between single and multiple past infections. Furthermore, the Anti-S antibody response was measured based solely on the receptor-binding domain (RBD), which may not fully represent the complete immune response.

## 5. Conclusions

By the time Ethiopia scaled up its vaccination campaign, the majority of the population had already been infected by the virus, indicating that the vaccination strategy did not significantly affect the pandemic course. Participants who had a previous SARS-CoV-2 infection and received one dose of the vaccine had better antibody responses and a lower rate of infection compared to those who had no evidence of previous SARS-CoV-2 infection. Our cost-effectiveness analysis also indicated that a one-dose vaccine approach in high seroprevalent settings with limited vaccine availability is an optimal strategy. Based on this evidence, the early implementation of a more targeted approach, tailored to the local context, could have resulted in a higher vaccination rate and potentially prevented later surges of the outbreak in Ethiopia.

## Figures and Tables

**Figure 1 vaccines-12-00745-f001:**
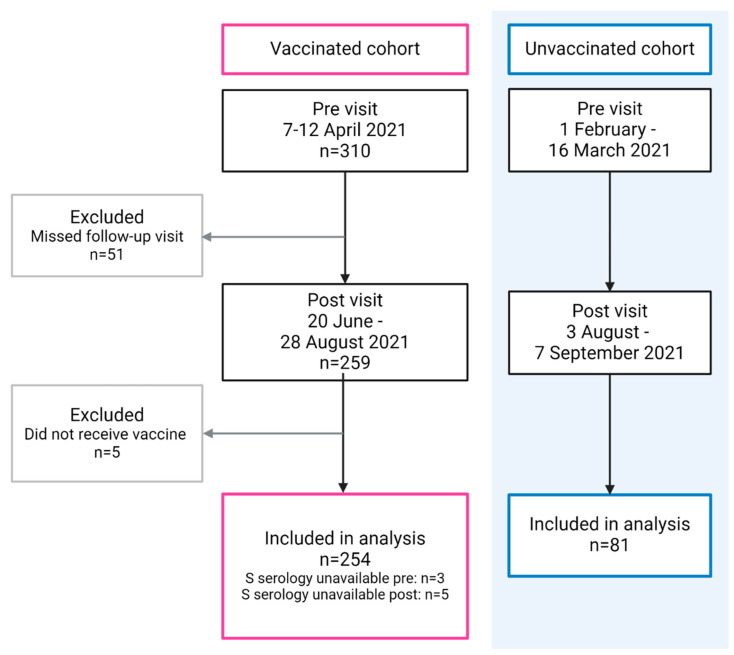
Flow diagram of participant recruitment at baseline (pre) and follow-up (post) visits.

**Figure 2 vaccines-12-00745-f002:**
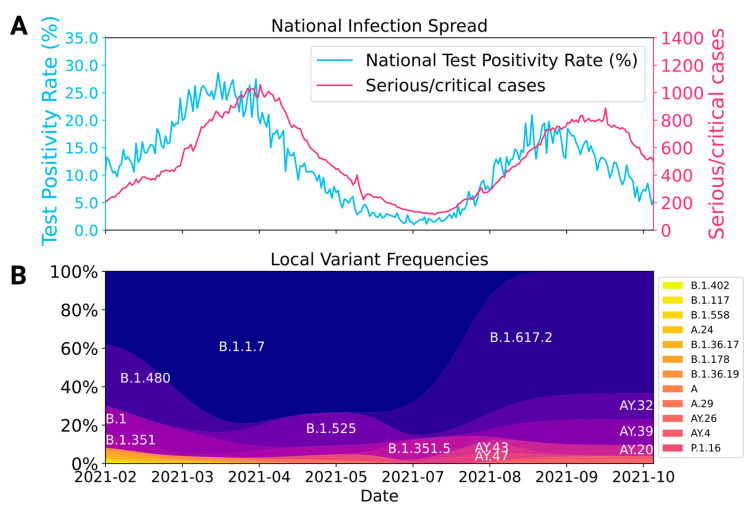
National infection spread and local SARS-CoV-2 variant pattern during the study period. (**A**) National RT-PCR test positivity rate and critical cases of COVID-19 in Ethiopia. (**B**) Frequencies of SARS-CoV-2 variants sequenced in Jimma based on 419 RT-PCR positive samples collected from January to September 2021 [24].

**Figure 3 vaccines-12-00745-f003:**
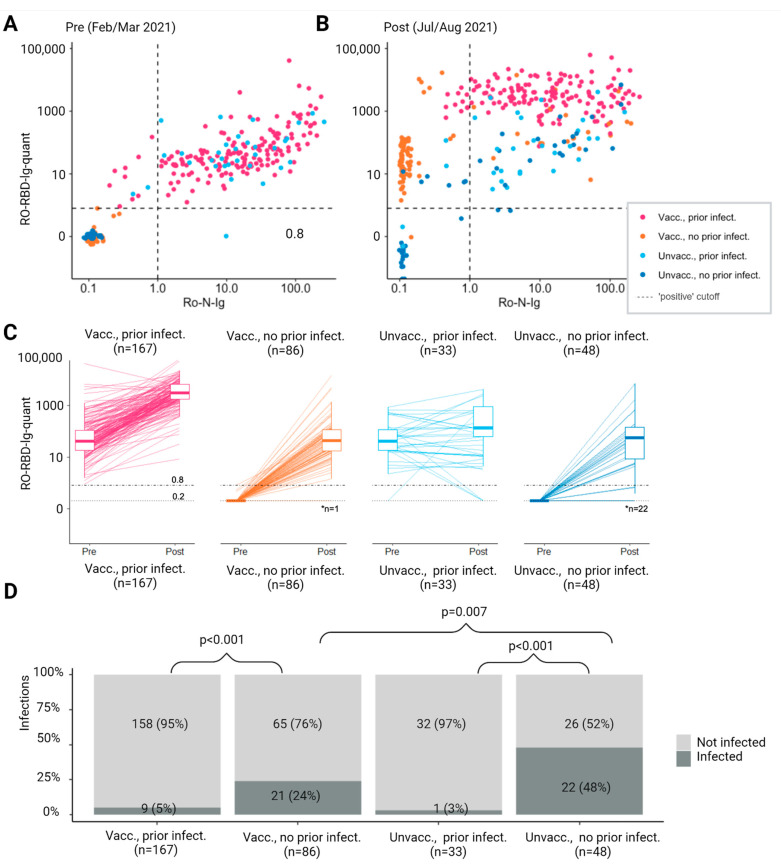
(**A**,**B**) Pre-post antibody response by vaccination status and evidence of prior SARS-CoV-2 infection. Participants with either N- or S- specific antibody response above the cutoff for positivity at the pre visit were considered to have prior infection. Relationship between N- and S-specific antibody response. (**C**) Individual change and summary boxplots in S-specific antibody response. (**D**) Frequency of infection according to vaccination status and prior infection. Notes: (**C**) Dotted line: Anti-S antibodies under 0.2 were categorized at 0.2. Dashed line: positive cutoff considered as evidence for receptor-binding domain-specific (RBD) antibodies. * Number of participants with titer below 0.2 (**D**) Only significant *p*-values are shown.

**Figure 4 vaccines-12-00745-f004:**
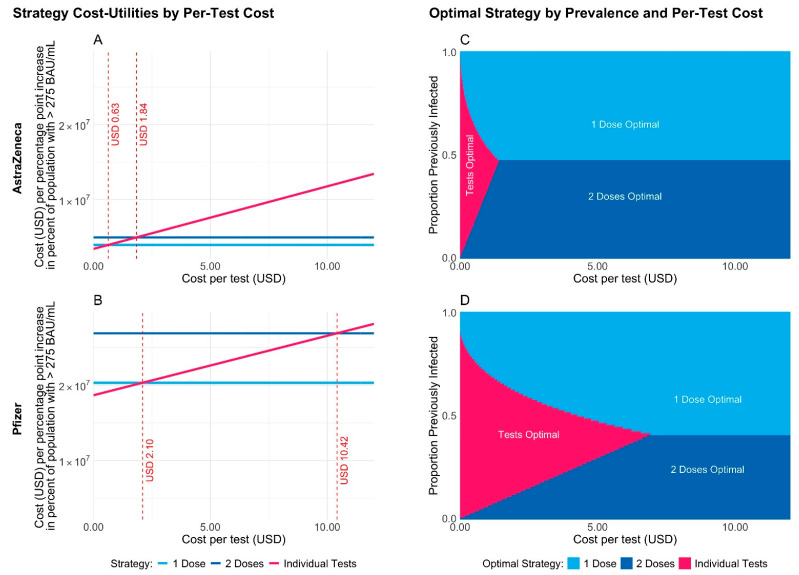
The cost in USD per unit increase in utility (i.e., per percentage point increase in the percentage of the adult population of Ethiopia with anti-spike levels above 275 BAU/mL compared to a no-vaccination scenario) for three different vaccination strategies and for AstraZeneca (**A**) vs. Pfizer (**B**). The strategy with the lowest cost per unit increase in utility for varying per-test costs and previous infection prevalence for AstraZeneca (**C**) vs. Pfizer (**D**).

**Figure 5 vaccines-12-00745-f005:**
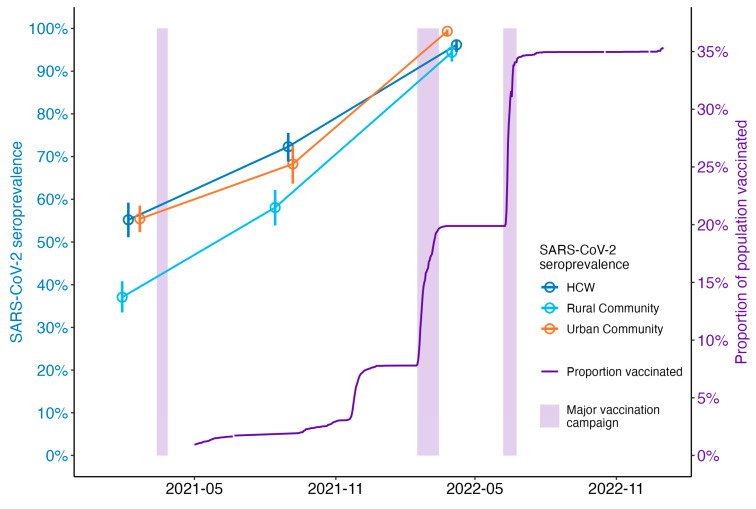
SARS-CoV-2 seroprevalence and proportion vaccinated. Seroprevalence was estimated among HCWs and general community members from a large seroepidemiological survey [24]. Population proportion vaccinated was estimated based on Ethiopian national data (https://www.facebook.com/EthiopiaFMoH, accessed on 26 December 2023) and [11]. Major vaccination campaigns in Ethiopia are indicated by shaded areas.

**Table 1 vaccines-12-00745-t001:** Baseline characteristics of study participants.

	Vaccinated Cohort (n = 254)	Unvaccinated Cohort (n = 81)
Participant category	Healthcare workers	Community
Pre-vaccination visit period	7–12 April 2021	1 February–16 March 2021
Sex, n (%)		
Male	174 (68.5)	42 (51.9)
Female	80 (31.5)	39 (48.1)
Age in years, mean (SD)	32.3 (7.9)	32.3 (12.3)
Pre-existing medical condition, n (%)	16 (6.3)	6 (7.4)
SARS-CoV-2 seroprevalence, n (%)		
By anti-nucleocapsid antibody	158 (62.2)	31 (38.5)
By anti-spike antibody	165 (65)	32 (39.5)

## Data Availability

Serological data supporting the interpretation of this manuscript will be available upon request to the corresponding author after publication. The variant sequences are published in the Sequence Read Archive under project number PRJNA1017685.

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
