# Peer review of "Tailoring COVID-19 Vaccination Strategies in High-Seroprevalence Settings: Insights from Ethiopia"

_vaccines, 2024, doi:10.3390/vaccines12070745_

Round 1

Reviewer 1 Report

Comments and Suggestions for Authors

This paper describes a study of the cost-effectiveness of various COVID-19  vaccination strategies in Ethiopia using interviews and serological tests data from healthcare workers and community participants. Overall, the study appears to be well done and the paper is well written. Here are a couple of items to attend to in a revision.

First, it would be good to label the two parts of Figure 2 as A and B which then are referred to in the title of the figure. 

Second, since the headline focus of the paper is on the cost-effectiveness findings, it would be good to add more details in Section 3.3 on how the cost-effectiveness part of the study was conducted instead of just referencing the Supplementary Material.

Author Response

Please see the attachment for response to reviewer's comments

Reviewer 2 Report

Comments and Suggestions for Authors

The paper assesses the cost-effectiveness of different COVID-19 vaccination strategies in Ethiopia, a high seroprevalence setting. The study involved healthcare workers (HCWs) and community participants and used interviews, serological tests, and national COVID-19 data to perform a cost-effectiveness analysis. The study suggests a single-dose vaccination strategy for seropositive individuals and a two-dose strategy for seronegative individuals as the most economical approach in such settings.

Strengths:

1. The study provides valuable insights tailored to Ethiopia's local context, which is crucial given the unique challenges faced by low-income settings.

2. The integration of serological tests, interviews, and national COVID-19 data offers a robust dataset for analysis.

3. The study highlights the concept of hybrid immunity (immunity from past infection and vaccination), which is critical for understanding vaccine strategy efficacy in high seroprevalence regions.

4. The emphasis on cost-effectiveness is highly relevant for resource-limited settings, providing actionable insights for policymakers.

Drawbacks:

1. Conducting only one antibody assessment three months after vaccination limits our understanding of long-term immunity and antibody durability.

2. The study lacks a longitudinal follow-up to observe the long-term impact of the vaccination strategies on infection rates and immunity.

3. The study is confined to a specific region (Jimma Medical Center) and may not fully represent the diverse epidemiological landscapes across Ethiopia.

4. The data used are a few years old, which might not reflect the current situation or the evolution of the pandemic and vaccination response.

5. Reliance on the Elecsys® anti-SARS-CoV-2 anti-nucleocapsid antibody test might have missed some infections due to signal loss over time, and it does not distinguish between single and multiple past infections.

Recommendations:

1. Implement longitudinal studies to track antibody levels and infection rates longer after vaccination. This will provide a clearer picture of the long-term efficacy of the vaccination strategies.

2. Include multiple regions within Ethiopia to ensure the findings represent the national context, considering regional variations in seroprevalence and healthcare infrastructure.

3. Utilize the most current data to ensure the analysis reflects the latest trends and challenges in the COVID-19 pandemic and vaccination efforts.

4. Use a combination of serological tests to distinguish between single and multiple infections and to improve the accuracy of detecting past infections.

5. Encourage local evidence generation and synthesis to guide policy, addressing the shortage of region-specific data that hampers effective decision-making.

6. Develop flexible vaccination strategies that can be adapted based on real-time seroprevalence data and local healthcare capabilities to maximize vaccine impact efficiently.

7. Enhance communication strategies to address vaccine hesitancy and misinformation, leveraging local data to build trust and encourage vaccination.

Author Response

Please see the attachment for response to reviewer's comments.

Reviewer 3 Report

Comments and Suggestions for Authors

The work is well planned and adequately executed. The authors convincingly showed that against the background of a previous history of COVID-19, the Covishield vaccine allows the formation of statistically significantly stronger adaptive immunity, even with a single dose. The latter is especially important in the context of an acute shortage of specific vaccines, which was observed in Ethiopia. An analysis of vaccinations carried out in Ethiopia clearly demonstrated that in conditions of shortages of vaccines and financial resources and against the background of high seropositivity due to natural disease, a single vaccination may be epidemiologically justified. We can only recommend checking the conclusions formulated on other segments of the Ethiopian population.

Author Response

(The authors gave the same response as above.)

Reviewer 4 Report

Comments and Suggestions for Authors

The manuscript describes cost effectiveness analysis of anti-COVID-19 vaccination in the area where pre-vaccination pandemic was eminent. The cost effectiveness was analyzed based on the cost of 2 vaccines (with one actually used) and varied antibody testing costs.

The results of analysis essentially valuable as one example in specific existing area and scenario.

The analytical strategy is cost analysis per percentage point increase in the percentage of antibody positive population.

The strong point of the research is original and important viewpoint in public health. The weakness of the research is retrospective (COVID-19 itself will never occur again) and specific (condition that testing is already available, vaccine arrival is after pandemic and vaccination budget is limited) nature limiting the implication of the research and utilization of cost analysis strategy, less persuasive than cost-utility analysis.

Suggestion:

1)In figure 1, both group should be presented in comparable manner as they will be compared in table 1, i.e., “included in analysis n=81 should be appear on the bottom of control group.”

The manuscript describes cost effectiveness analysis of anti-COVID-19 vaccination in the area where pre-vaccination pandemic was eminent. The cost effectiveness was analyzed based on the cost of 2 vaccines (with one actually used) and varied antibody testing costs.

The results of analysis essentially valuable as one example in specific existing area and scenario.

The analytical strategy is cost analysis per percentage point increase in the percentage of antibody positive population.

The strong point of the research is original and important viewpoint in public health. The weakness of the research is retrospective (COVID-19 itself will never occur again) and specific (condition that testing is already available, vaccine arrival is after pandemic and vaccination budget is limited) nature limiting the implication of the research and utilization of cost analysis strategy, less persuasive than cost-utility analysis.

Suggestion:

1)In figure 1, both group should be presented in comparable manner as they will be compared in table 1, i.e., “included in analysis n=81 should be appear on the bottom of control group.”

Author Response

(The authors gave the same response as above.)
